# Factors Influencing Patients Using Long-Term Care Service of Discharge Planning by Andersen Behavioral Model: A Hospital-Based Cross-Sectional Study in Eastern Taiwan

**DOI:** 10.3390/ijerph18062949

**Published:** 2021-03-13

**Authors:** Yi-Chien Chen, Wei-Ting Chang, Chin-Yu Huang, Peng-Lin Tseng, Chao-Hsien Lee

**Affiliations:** 1Department of Nursing, Taipei Veterans General Hospital Yuli Branch, Hualien 981002, Taiwan; yichien0606@gmail.com (Y.-C.C.); j5053@mail.vhyl.gov.tw (C.-Y.H.); 2Department of Nursing, Meiho University, Pingtung 912009, Taiwan; yahapply@gmail.com; 3Department of Family Medicine, Lotung Poh-Ai Hospital, Yilan 265501, Taiwan; arieltin@hotmail.com; 4Department of Nursing, Pingtung Christian Hospital, Pingtung 900026, Taiwan; 5Department of Health Business Administration, Meiho University, Pingtung 912009, Taiwan

**Keywords:** discharge planning, long-term care, Taiwan

## Abstract

Taiwan has been an aged society since March 2018, and the elderly population suffer from multiple comorbidities and long duration of disability. Therefore, the service of discharge planning of long-term care 2.0 is an important stage before patients go back to the community. Strengthening the sensitivity when identifying predisabled patients is a principal development of discharge planning. In the current study, we analyzed the characteristics and predictive factors of patients who used the service of long-term care 2.0 from the perspective of discharge planning. In this retrospective study, we included patients who received the discharge planning service in a hospital located in southern Hualien during November 2017 to October 2018. The data were collected and classified as predisposing factors, enabling factors, and need factors according to the analysis architecture of the Andersen Behavioral Model. There were 280 valid patients included in this current study; age, medical accessibility, possession of a disability card, and cerebrovascular diseases, cardiovascular diseases, and diabetes mellitus were the vital factors which influenced the coherence and cohesion between discharge planning and the service of long-term care 2.0. Among them, the most influencing factor was age. We hope that the current study will make policymakers in hospitals pay attention to the usage of the discharge planning service to link long-term care 2.0 and effectively promote the usage of long-term care 2.0.

## 1. Introduction

Taiwan has been an aged society since March 2018. Until January 2021, the population older than 65 years old was 16.15% and those older than 85 years old equated to 10.3% (total population in Taiwan was 23,548,633) [1]. According to the report of World Bank (2019), Japan (27%), Italy (23%) and Germany (21%) were the countries with the most severe aging populations [2]. In 2015, the World Bank East Asia and Pacific Regional Reports disclosed that the population in east Asia and the Pacific Ocean were rapidly aging [3], including Taiwan. The population older than 65 years old in Taiwan was 7.1% in 1993, rising up to 14.1% in 2018. It only took 25 years for Taiwan to reach the standard of an aged society (7% to 14%) as defined by the WHO, which is as quick as the aging rate in Japan (24 years). It took Europe and America at least 60 to 100 years to reach the standard of an aged society [4]. It is predicted that we may step into super-aged society in 2025, and Taiwan’s elderly population may rank second in the world, exceeding Korea, Japan and Hong Kong in 2060 (the country ranked first is predicted to be Qatar) [5]. Therefore, in response to the development of an aged society, long-term care has been regarded as an important policy since 2007, and a series of plans have been vigorously promoted, including the connection between the discharge planning and long-term care 2.0. The difference between long-term care in Taiwan and other countries is that the former have national health insurance, which is a compulsory insurance policy and adopts a nationally consistent rate range. The national health insurance can also provide medical protection for economically disadvantaged groups. According to the global population structure, the number of people who need the service of long-term care will increase year by year.

Long-term care in Taiwan began with discharge planning in 1994, which is a centralized, coordinated and continuous process of collaboration with patients and their families through interprofessional medical teams [6]. Moreover, discharge planning also emphasizes the connection of long-term care resources in the community during hospitalization, which is an important preparation for patients to return to the community. National Ten-year Long-term Care Plan 1.0 (long-term care 1.0) was performed during 2007 to 2016 to promote local aging, which included the service of respite care at home, home care, home nursing care, home rehabilitation, transportation, assistive device service, nutritional catering and institutional services [7]. However, there were many limitations of long-term care 1.0. Only those people in the following categories could use long-term care 1.0: those with a disability card for and older than 50 years old, mountain aboriginals who were older than 55 years old, the disabled and those older than 65 years old, and those who are dependence on instrumental daily life activity and older than 65 years old [8]. The payment of the long-term care service was inflexible—the service hour of different item was fixed, and the payment of service could not be cross-project. Additionally, the lack of care attendants and overworking led to restrictions of service amounts and low retention rates (24%). Therefore, long-term care 1.0 was upgraded to 2.0 in 2017. The served population expanded to fragile elders older than 65 years old, people younger than 49 years old with a disability card, people older than 50 years old with dementia, and disabled aboriginals who are older than 55. After the expansion, the number of service recipients increased from 511,000 to 738,000, with an increase of about 44% [8]. Moreover, the service items also expanded from 8 to 17, extending to the prevention stage and to the terminal stage. The newly included items are: dementia day-care, community integration, caregiver supporting bases, community prevention care, prevention/delay disability, discharge planning, and home medical care. the Long-term care was promoted in several ways so that people can understand how to access. In addition to integrating the sanitation and social welfare, long-term care 2.0 also extends the recipients, establishes a standard operation procedure, builds a shared information platform for care service management and a communication bridge between the hospital and sanitation, and uses the media to promote the people’s cognition of long-term care 2.0 [9].

The average duration of long-term care for a Taiwanese person in need of it is 7.3 years, because as the population in Taiwan became older, comorbidities and disability increase [10]. The average age for receiving discharge planning before long-term care was between 63 and 70 years old [11,12,13]. Discharge planning is a way to connect returning patients to the community as it can identify the patients who may potentially develop disabilities, shorten the duration of hospitalization, and decrease the possibility of readmission [14]. According to a review in 2017, patients referred to long-term care 2.0 through discharge planning had reduced durations (5.85 days) of care than patients who received long-term care after discharge (from 17.4 to 11.55 days), and the number of people served from 113,706 to 124,544 was increased [14].

There are several studies on discharge planning showing that it could shorten the duration of hospitalization, reduce readmission rate, lower the medical cost, and raise the medical care quality [12,13,15]. However, research on the connection between discharge planning and long-term care is scanty. Therefore, it is worth exploring the predisposing factors that influence the utilization of long-term care [16]. Therefore, the purpose of the current study, is to uncover the factors affecting patients who have already received the service of discharge planning and are willing to accept the service of long-term care, and promote the experience of long-term care in Taiwan and share it with other countries.

## 2. Methods

### 2.1. Background Information

In this cross-sectional study, a case review was used, and the patients who received the service of discharge planning at the hospital (Taipei Veterans General Hospital Yuli Branch), which is located in the southern Hualien, were included during November 2017 to October 2018. The number of beds in the hospital was 95. This current study passed the audit of the Institutional Review Board, and the trial number was 19-008-C. In addition to receiving the service of discharge planning, the patients who were included matched the criteria for long-term care 2.0: persons younger than 49 years old with card of disability, persons older than 50 years old with dementia, disabled mountain aboriginals who are older than 55 years old, persons older than 65 years old who conduct daily dependent activities or persons older than 65 years old living alone who carry out fundamental daily activities dependently. In total, 294 patients were included and received the service of discharge planning during hospitalization. The excluded criteria were: expire during hospitalization, transferal to other hospital, or not compatible with the conditions of long-term care 2.0. However, 14 patients were excluded (6 patients expired, 5 patients were not suitable with the rule of long-term care 2.0, and 3 patients were transferred to other hospital), and there were 280 patients included finally. We asked 280 patients to fill out a brief questionnaire after receiving the service of discharge planning during the hospitalization (Appendix A).

### 2.2. Data Collection

According to the Andersen Behavioral Model, personal medical behavior is influenced by predisposing factors, enabling factors and need factors [17]. Moreover, in recent decades, the Andersen Behavior Model has been widely applied in a large amount of literature on the discussion of medical services and other associated factors. Therefore, we designed the structure of the current study based on the Andersen Behavioral Model [17]. We collected data on the reasons disabled elderly people needed care in Taiwan, Japan and the United States of America, and separated it into predisposing factors, enabling factors and need factors [18,19,20]. Predisposing factors included age and sex; enabling factors included medical accessibility (the area in our study is narrow and long, the length is around 100 km, and the medical resources are scanty); need factors included tubal insertion (nasogastric tube, Foley tube, tracheostomy, colostomy, etc.), card for person with disability, chronic diseases (cerebrovascular disease, dementia, osteoarthritis, coronary/cardiovascular disease, chronic obstructive pulmonary disease/asthma, malignancy, end stage renal disease, diabetic mellitus). From November 2017 to October 2018, in Taipei Veterans General Hospital Yuli Branch, we recruited 280 valid patients who were compatible with the conditions of long-term care 2.0 and invited them for a standardized interview.

### 2.3. Statistical Analysis

Patients’ characteristics were described by absolute and relative frequencies for categorical variables and means and standard deviations for continuous variables. For the inference statistic methods, a Chi-square test was used to analyze the difference in the proportion of categorical variables between the patients with and without using long-term care service of discharge planning. Additionally, the significant factors in each statistic testing were considered in logistic regression model, and the optimal model was determined by stepwise model selection. These results were reported as odds ratios (ORs) with 95% confidence interval (CIs). A *p*-value < 0.05 was considered to be statistically significant. Statistical analysis was performed using IBM SPSS Statistics 23.

## 3. Results

### 3.1. Characteristic of the Patients Who Receiving the Discharge Planning Service

In this study, patients’ average age was 73.7 ± 13.3 years old, 77.5% patients were over 65 years old, and only around 5.7% of patients were aged below 50 years old (Table 1). It revealed that patients who predominantly use the service of discharge planning are the elderly and males. Only 43.2% patients lived near the hospital, which means the southern Hualien was short of medical resources. In total, 26.8% of patients required catheters (Foley, nasogastric tube, or tracheostomy) and 67.1% of patients were without disability cards. The top three chronic diseases were diabetes mellitus (35.4%), followed by bone/joint diseases and respiratory diseases (asthma, chronic obstructive pulmonary disease) (33.6%), and coronary arterial disease/cardiovascular diseases (27.5%).

### 3.2. The Influencing Factors Affecting Patients Who Received the Discharge Planning Service of Long-Term Care 2.0

We analyzed the influencing factors which affected patients who had received the discharge planning service of long-term care 2.0 using the Andresen Behavioral Model (Table 2). Based on the age distribution, we could understand that most family was elderly couples whether long-term care used or not. Especially, there were up to 65.7% patients, the mean age 75.1 ± 10.9, using long-term care resources, and 71.0% patients, who were older than 65 years old, used long-term care resources after discharge. It was revealed that age (*p* = 0.001) was a significant factor for use of long-term care 2.0. In terms of enabling factors, 58.5% patients who used long-term care resources lived in areas without medical institutions, which emphasizes that a medical institution in the place residences would make a significant difference (*p* = 0.004). In terms of need factors, a card for a person with disability affects whether that patient uses long-term care after discharge or not (*p* = 0.002). In the past, a significant difference of the long-term care 2.0 utilization in cerebral vascular diseases/stroke (*p* = 0.002), coronary arterial disease/cardiovascular disease (*p* = 0.003) and diabetes mellitus (*p* = 0.009) has been shown. According to these results, the significant predictive factors for using the service of long-term care included age, medical accessibility, cards for persons with disability, cerebrovascular disease, coronary or cardiovascular disease and diabetic mellitus. However, there was no significant association between sex, tubal insertion, dementia, osteoarthritis, malignancy or end stage renal disease and the utilization of long-term care 2.0.

### 3.3. The Factors and Degree of Influence for Connecting Patients from Discharge Planning Service to Long-Term Care 2.0

Patients who were over 65 years old vs. younger than 50 years old (OR: 9.773, 95%; CI: 2.752–34.705, *p* < 0.001) or 50–65 years old (OR: 3.991, 95%; CI: 1.031–15.452, *p* = 0.045) vs. younger than 50 years old were found to be at increased risk of being transferred to long-term care 2.0. In terms of enabling factors, the patients with medical accessibility (OR: 2.300, 95%; CI: 1.290–4.103, *p* = 0.005) were at an increased risk of being transferred to long-term care 2.0. In terms of need factors, those with disability cards (OR: 2.925, 95%; CI: 1.514–5.652, *p* = 0.001) or cerebrovascular disease/stroke (OR: 2.357, 95%; CI: 1.095–5.076, *p* = 0.028), coronary/cardiovascular disease (OR: 2.526, 95%; CI: 1.300–4.908, *p* = 0.006) or diabetic mellitus (OR: 2.191, 95%; CI: 1.172–4.097, *p* = 0.014) faced an increased risk of being transferred to long-term care 2.0.

## 4. Discussion

### 4.1. Characteristic of the Patients Who Received the Discharge Planning Service

In the clinical experience, it was found that many patients had preclinical symptoms before joining the long-term care system, which means that the long-term care should extend to the discharge planning service for early intervention. Moreover, in this current study, more than 60% patients who received the service of discharge planning would utilize the resource of long-term care 2.0. The average age of patients was 73.7 years old, which was equal to the patient in the city in Taiwan. According to other studies, more than 70% patients were older than 65 years old [11,12,13]. Obviously, aging populations are a global problem, not just in Taiwan. The caring burden of young adults is becoming larger and larger, which may have a great impact on the overall economic capacity. Therefore, the long-term care system will help diminish the dilemma. The average life expectancy in Taiwan is 77.3 years old for men and 83.7 years old for women [5]. Different from other research, the patients who were included in our study were predominantly male [20,21]. It was speculated that the average life expectancy of men in Taiwan was less than that of women, and it was possible that men might suffer from disability and need long-term care earlier. The accessibility of medical services has been an important consideration in the allocation of healthcare resources in Taiwan [18,22]. As high as 56.8% of patients lived in places with no medical institutions. For elderly couples who lived without children, the transportation distance would affect the willingness of seeking medical help early and the motivation to be discharged. The current study also highlights the importance of an effective acquisition of long-term care resources in medically underserved areas. Not fulfilling a elderly patients’ medical needs after discharge would increase the usage of emergency rooms and unplanned return outpatient clinic visits or hospitalizations [23]. Therefore, confirming the long-term care resources before discharge and ensuring that resources are obtained as soon as possible after discharge are the essence of the discharge planning service, connecting to long-term care 2.0. In terms of need factors, only 26.8% patients underwent tubal insertions. In the amendment of the policy, it was indicated that tubal insertion was no longer a prominent feature of screening high demand patients for the discharge planning service. Because our culture and population structure are similar to those of Japan, we may use Japan as a benchmark for long-term care and develop our own “Nursing Care and Prevention Screening Checklist”; this would enable front-line case managers of the discharge planning service to follow and improve the sensitivity of excavating patients [19].

Those with cards for disabilities represented the threshold affecting the usage of long-term care 2.0 for patients under 49 years old in Taiwan [9]. Nevertheless, up to 70% of patients who needed the assistance of long-term care were not equipped with the card for persons with disabilities, and the age of onset of cancer, the leading cause of death, had become younger in recent years [24]. Along with the progression of cancer, the activity of daily life (ADL) of the patients would be more dependent and increases the demand of long-term care. A number of cancer patients were younger than 49 years old and were not eligible for a card for persons with disabilities. In addition, cancer was also listed as one of the main causes of disability and demands of long-term care in Japan (1.9%) [19]. Therefore, using a card for persons with disabilities to set the threshold of applicability of long-term care must be the subject of further discussion and amendment.

In terms of chronic diseases, first, diabetes continues to be a disease affecting many civilizations and was one of the top ten causes of death in Taiwan—nearly 10,000 people die each year due to diabetes [24]. Therefore, diabetes was also the target of chronic disease prevention and discharge planning service screening [25]. Second, chronic obstructive pulmonary disease (COPD) and cerebrovascular disease were also the cause of unplanned rehospitalization within 14 days after discharge. The incidence or death rate is increasing [26]. We recommended that those with chronic diseases should be listed as a high risk group for discharge planning service screening in Taiwan to improve the sensitivity of those who need long-term care 2.0.

### 4.2. The Influencing Factors Affecting That Patients Who Had Received the Discharge Planning Service Utilized Long-Term Care 2.0

Another study revealed that 35.0% of elderly people had lower ADL after discharge [27]. This highlights the importance of screening the patients among the discharge planning service to connect with long-term care 2.0, so as to prevent not including patients who are in need of nursing care or medical help after discharge. In our study, it was found that predisposing factors, enabling factors, and need factors all affected the usage of long-term care 2.0. The same outcomes as several studies were found—i.e., that age, those with cards for persons with disabilities, and past history of cerebrovascular disease and coronary arterial disease were the significant factors affecting the usage of long-term care [28]. There has been no study exploring whether medical accessibility and diabetes might influence the connection between the discharge planning service and long-term care 2.0. Therefore, an important result in our study is that a significant difference regarding using long-term care resources among diabetic patients and the place of residence without medical institutions nearby was found.

In the area of this study, most families were elderly couples, and the research hospital was located in a rural area of eastern Taiwan with a lack of medical resources. Due to the inconvenient transportation, the willingness to use long-term care resources might be affected. Therefore, we suggested that further qualitative research should be designed to discuss the association between the use of long-term care 2.0 and medical accessibility.

Additionally, due to the irreversibility of diabetes, diabetic patients had an average 10-year lifespan less than other patients [29]. In other words, the functional performance would decrease and the complications would increase in diabetic patients as time goes by, and the ADL would be dependent and aggravated gradually [25]. In this current study, the results are bound to make up for the gaps in past research and made the important points for the further study.

### 4.3. The Factors and Degree of Influence for Connecting Patients from Discharge Planning Service to Long-Term Care 2.0

In the research of Mitchell and Krout (1998), it was revealed that predisposing factors and enabling factors had weak effect on the behavior of medical service, while need factors were the most important predictors [30]. The outcome was different from our study. We analyzed the factors and degree of impact for connecting patients from discharge planning service to long-term care 2.0 by logistic regression. The statistical significance was 0.05 and the results are disclosed by Table 3. Age, as one of the predisposing factors, was the most dramatic factor. In our study, the most influential factors regarding the connection between discharge planning service and long-term care resources were predisposing factors, and this meant that age was the main factor affecting the usage of long-term care. On the contrary, age might also be the resistance to using long-term care in cancer patients.

In terms of need factors, having a disability card was the most influential factor, which was the same as the research of Huan-Yui Tseng and Yang Shin [21]. However, if the patient’s age does not reach the threshold and they have no disability card, they may not be allowed to use the resources of long-term care 2.0. For borderline patients, such as young advanced cancer patients, the policymakers should pay attention to amending the criteria for entering the gate of long-term care 2.0.

In chronic diseases, patients with coronary arterial/cardiovascular diseases had greater chance of using long-term care than the patients with cerebrovascular diseases. This result subverted previous clinical experience. It might be speculated that the patients with coronary arterial/cardiovascular diseases might also have had other comorbidities, and the functional performance would be influenced by disease progression. Further study to analyze the correlation between patients with these two chronic diseases and the usage of long-term care is recommended.

## 5. Study Limitation

In this study, we only included the patients who received the discharge planning service at the hospital in southern Hualien, but we did not include the main caregivers. However, the main caregivers played an important role in patients’ care and the usage of long-term care resources, and they were often neglected in the policymaking [31]. Moreover, age was the most important factor affecting the usage of long-term care. In further studies, we suggest expanding the study group and using a cohort study to analyze the connection from discharge planning service to the characteristics of patients who receive long-term care and the caregivers by a cohort study in the future in Taiwan. Tracking the changes in the course of disease and the usage of medical services should be used as bases for contingency use of long-term medical services.

## 6. Conclusions

In this current study, it was revealed that predisposing factors, enabling factors, and need factors all affected the usage of long-term care 2.0 in patients who received the discharge planning service. Among them, the predisposing factor of age was the most influential. With the aging of the population and the increase in the complication of diseases, the demand of long-term care in Taiwan will only grow in the future. Nevertheless, the resources of long-term care are not only for the elderly but also for patients with multiple comorbidities and catastrophic illnesses. In Taiwan, some disabled patients who do not meet the criteria for a disability card, are too young for long-term care 2.0, which leads to the dilemma of patient care after being discharged. It is necessary to be more cautious in policymaking to perfect care in the future. We may recommend that the policymaker develop other functional scales to evaluate young disabled people. For example, scales for performance status, such as Eastern Cooperative Oncology Group (ECOG) Performance Status or Karnofasky Performance scale, for young advanced cancer patients are more suitable for evaluating standard of using long-term care 2.0.

## Figures and Tables

**Table 1 ijerph-18-02949-t001:** Characteristics of patients using service of discharge planning (N = 280).

Variable	N (%)
**Predisposing factors**	
Age (*M* ± *S.D*)	73.7 ± 13.3
<50 years	16 (5.7)
50–65 years	47 (16.8)
>65 years	217 (77.5)
Sex	
Male	160 (57.1)
Female	120 (42.9)
**Enabling factors**	
Medical accessibility	
Yes	121 (43.2)
No	159 (56.8)
**Need factors**	
Tubal insertion	
Yes	75 (26.8)
No	205 (73.2)
Card for person with disability	
Yes	92 (32.9)
No	188 (67.1)
Chronic diseases	
Cerebrovascular disease	61 (21.8)
Dementia	27 (9.6)
Osteoarthritis	94 (33.6)
Coronary or cardiovascular disease	77 (27.5)
Chronic obstructive pulmonary or asthma	94 (33.6)
Malignancy	22 (7.9)
End stage renal disease	18 (6.4)
Diabetic mellitus	99 (35.4)

**Table 2 ijerph-18-02949-t002:** Characteristics of patients by utilization of long-term care 2.0 (N = 280).

Factors			LTC	No LTC	*X* ^2^	*p*-Value
Total			184 (65.7)	96 (34.3)		
Predisposing	Age (M ± S.D)		75.1 ± 10.9	70.8 ± 16.5	14.364 **	0.001
		<50 years	5 (31.3)	11 (68.8)		
		50–65 years	25 (53.2)	22 (46.8)		
		>65 years	154 (71.0)	63 (29.0)		
	Sex	Male	105 (65.6)	55 (34.4)	0.001	0.971
		Female	79 (65.8)	41 (34.2)		
Enabling	Medical accessibility	Yes	91 (75.8)	30 (24.2)	8.522	0.004
		No	93 (58.5)	66 (41.5)		
Need	Tubal insertion	Yes	49 (65.3)	26 (34.7)	0.007	0.935
		No	135 (65.9)	70 (34.1)		
	Card for person with disability	Yes	72 (78.3)	20 (21.7)	9.573 **	0.002
		No	112 (59.6)	76 (40.4)		
	*Chronic diseases*					
	Cerebrovascular disease	Yes	50 (82.0)	11 (18.0)	9.144 **	0.002
		No	134 (61.2)	85 (38.8)		
	Dementia	Yes	19 (70.4)	8 (29.6)	0.288	0.592
		No	165 (65.2)	88 (34.8)		
	Osteoarthritis	Yes	62 (66.0)	32 (34.0)	0.004	0.951
		No	122 (65.6)	64 (34.4)		
	Coronary or cardiovascular disease	Yes	61 (79.2)	16 (20.8)	8.599 **	0.003
		No	123 (60.6)	80 (39.4)		
	Chronic obstructive pulmonary or asthma	Yes	67 (71.3)	27 (28.7)	1.943	0.163
		No	117 (62.9)	69 (37.1)		
	Malignancy	Yes	16 (72.7)	6 (27.3)	0.521	0.470
		No	168 (65.1)	90 (34.9)		
	End stage renal disease	Yes	13 (72.2)	5 (27.8)	0.362	0.548
		No	171 (65.3)	91 (34.7)		
	Diabetic mellitus	Yes	75 (75.8)	24 (24.2)	6.856 **	0.009
		No	109 (60.2)	72 (39.8)		

LTC—long-term care; **—*p*-value < 0.01.

**Table 3 ijerph-18-02949-t003:** Logistic regression model for factors of discharge planning on referrals to long-term care 2.0 (N = 280).

Factors		OR	95% CI	*p*-Value
Predisposing	Age			
	50–65 years vs. <50 years	3.991 *	[1.031, 15.452]	0.045
	>65 years vs. <50 years	9.773 ***	[2.752, 34.705]	<0.001
Enabling	Medical accessibility			
	Yes vs. No	2.300 **	[1.290, 4.103]	0.005
Need	Card for person with disability			
	Yes vs. No	2.925 **	[1.514, 5.652]	0.001
	Cerebrovascular Accident			
	Yes vs. No	2.357 *	[1.095, 5.076]	0.028
	Coronary/cardiovascular disease			
	Yes vs. No	2.526 **	[1.300, 4.908]	0.006
	Diabetic mellitus			
	Yes vs. No	2.191 *	[1.172, 4.097]	0.014

OR—Odds Ratio; CI—Confidence Interval; *—*p*-value < 0.05; **—*p*-value < 0.01; ***—*p*-value < 0.001.

## Data Availability

Not applicable.

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
