# Peer review of "Factors Influencing Patients Using Long-Term Care Service of Discharge Planning by Andersen Behavioral Model: A Hospital-Based Cross-Sectional Study in Eastern Taiwan"

_ijerph, 2021, doi:10.3390/ijerph18062949_

Round 1
Reviewer 1 Report
Dear Authors,
Thank you for giving me possibility to review of the presented paper. The topic of manuscript is relevant from public health perspective and it is remmended to present it. However, this paper needs some minor improvements:
Please use people with disability, elderly with disability, card for people with disability. It is recommended to include people in the centre, than to describe their impairment.
In the results section one mistake is shown in numbers: 76.1%, but in table 1 it is 67.1.
Results 3.3.
Please be more objective in this section. It is recommended not to judge. Please avoid statement “dramatic factors” in this section, but please include it in the discussion.
Discussion 4.2.
Please avoid “In Table 2”. This is reserved for result section.
In the second paragraph, you mentioned that most family was elderly couples. I do not see this description in the result section. Please complete this description in the result section.
Conclusion:
Please specify action plan for policy makers. It is necessary to present clear recommendation for policy makers: what to do, how to improve and provide a perfect care for some group of patients in your country. Please give universal recommendations for practicioners from the international perspectives.
Please mention about the need of developing the qualitative research among patients and medical staff. This is necessary to find the needs and to provide people -centre approach in the medical practice.
Author Response
Responses to Revision Request
REVIEWER #1:
Dear Authors,
Thank you for giving me possibility to review of the presented paper. The topic of manuscript is relevant from public health perspective and it is remmended to present it. However, this paper needs some minor improvements:
Please use people with disability, elderly with disability, card for people with disability. It is recommended to include people in the centre, than to describe their impairment.
We agree with the reviewer and revised the article base on your suggestion. We had fixed the presentation style of the article. Instead of “Disability Card”, it should be “card for person with disability”.
In the results section one mistake is shown in numbers: 76.1%, but in table 1 it is 67.1.
Thank you for these observations. We had fixed the presentation style of the article. Instead of “76.1%”, it should be “67.1%”.
Results 3.3.
Please be more objective in this section. It is recommended not to judge. Please avoid statement “dramatic factors” in this section, but please include it in the discussion.
Thank you for these observations. We had fixed the presentation style of the article. We deleted the statements “We analyzed the factors and degree of impact for connecting patients from discharge planning service to long term care 2.0 by Logistic regression. The statistical significance was 0.05, and the result was disclosed by Table 3. The age, in the predisposing factors, was the most dramatic factors.”.
Discussion 4.2.
Please avoid “In Table 2”. This is reserved for result section.
Thank you for these observations. We had fixed the presentation style of the article. Instead of “In Table 2”, it should be “In our study”.
In the second paragraph, you mentioned that most family was elderly couples. I do not see this description in the result section. Please complete this description in the result section.
Thank you for these observations. We had completed this description in the result section 3.2 as the statements, “Based on the age distribution, we could understand that most family was elderly couples whether long-term care used or not.”.
Conclusion:
Please specify action plan for policy makers. It is necessary to present clear recommendation for policy makers: what to do, how to improve and provide a perfect care for some group of patients in your country. Please give universal recommendations for practicioners from the international perspectives.
Thank you for these observations. We had completed this description in introduction as the statements, “The difference of long-term care between Taiwan and other countries is that we have our own national health insurance, which is a compulsory insurance policy and adopts a nationally consistent rate range. The national health insurance can also provide medical protection for economically disadvantaged groups. According to the global population structure, the number of people who need the service of long-term care will increase year by year.”.
Please mention about the need of developing the qualitative research among patients and medical staff. This is necessary to find the needs and to provide people -centre approach in the medical practice.
Thank you for these observations. We had completed this description in introduction as the statements, “Therefore, in this current study, we look forward to disclosing the causes which affect the patients who had already received the service of discharge planning willing to accept the service of long-term care, and wish to have the opportunity to promote the experience of long-term care in Taiwan and share it with other countries.”.

Reviewer 2 Report
This study analyzed the influencing factors of the service of long term care 2.0 utilization among patients who involved in the discharge planning by logistic regression. In this study, 280 patients were included. Three factors—predisposing factors, enabling factors and need factors— were adapted based on the Andersen Behavioral Model. The influencing factors on the service of long term care 2.0 utilization are well examined in this study—age, medical accessibility, disability card, and cerebrovascular diseases, cardiovascular diseases, and diabetes mellitus were the vital important factors. The results are applicable. This study has both theoretical and practical significance.
However, this manuscript would be improved if some problems could get addressed. Below are some comments/questions and suggestions, personally.
- The concepts of discharge planning and long term care are vaguely delineated. For example, the first and second paragraph on page 2 is not clear to readers. I think more detailed interpretation can make the concepts clearer by reviewing more literatures and policies.
- What are the specific differences between long term care 1.0 and 2.0? The author may know them well, but not the readers.
- Due to table 1, number “20” should be changed into “5.7” in line 20, page 3.
- Number “76.1” should be converted into “67.1” in line 24, page 3.
- Number “58.8” should be converted into “58.5” in line 3, page 4.
- The results of univariate analysis can only indicate whether the distribution difference between using long term care group and not using long term care group is statistically significant. In 3.2 section, the description of analysis result should be further modified.
- Word “men” should be changed into “mean” in line 7, page 5.
- In discussion section, the author stated “there is no difference between rural and urban area in our study”. The author need to show the analysis results in result section.
- In discussion section, the author may need to explain more why these factors could influence the usage of long term care and it would be great to compare with other research.
Author Response
REVIEWER #2
This study analyzed the influencing factors of the service of long term care 2.0 utilization among patients who involved in the discharge planning by logistic regression. In this study, 280 patients were included. Three factors—predisposing factors, enabling factors and need factors— were adapted based on the Andersen Behavioral Model. The influencing factors on the service of long term care 2.0 utilization are well examined in this study—age, medical accessibility, disability card, and cerebrovascular diseases, cardiovascular diseases, and diabetes mellitus were the vital important factors. The results are applicable. This study has both theoretical and practical significance.
However, this manuscript would be improved if some problems could get addressed. Below are some comments/questions and suggestions, personally.
The concepts of discharge planning and long term care are vaguely delineated. For example, the first and second paragraph on page 2 is not clear to readers. I think more detailed interpretation can make the concepts clearer by reviewing more literatures and policies.
What are the specific differences between long term care 1.0 and 2.0? The author may know them well, but not the readers.
Thank you for these observations. We had completed this description in introduction as the statements, “Long-term care in Taiwan begins at discharge planning in 1994, which is a centralized, coordinated and continuous process of collaboration with patient and their family through inter-professional medical teams [6]. Moreover, discharge planning also emphasizes the connection of long-term care resources in the community during hospitalization, which is an important preparation for patients to return to the community. National Ten-year Long-term Care Plan 1.0 (long-term care 1.0) was performed during 2007 to 2016 for promoting local aging, which included the service of respite care at home, home care, home nursing care, home rehabilitation, transportation, assistive device service, nutritional catering and institutional service [7]. However, there were many limitation of long-term care 1.0. Only those people with the following qualifications could use long-term care 1.0: card for people with disability and older than 50-year-old, mountain aboriginal who were older than 55-year-old, the disable and older than 65-year-old elderly, IADL-dependent and older than 65-year-old [8]. The payment of long-term care service was inflexible: the service hour of different item was fixed, and the payment of service could not be cross-project. Besides, the lack of care attendants and over working led to restrictions of service amounts and low retention rate (24%). Therefore, long-term care 1.0 was upgraded to 2.0 since 2017. The served population are expanded to fragile elder older than 65 years old, people younger than 49 years old with card of disability, people older than 50 years old with dementia, and disable aboriginals who are older than 55. After the expansion, the number of service recipients increased from 511,000 to 738,000, with an increase of about 44% [8]. Moreover, the content of service items also expands from 8 to 17, extending forward to the prevention stage and backward to the terminal stage. The newly included items are: dementia day-care, community integration, caregiver supporting bases, community prevention care, prevention/delay disability, discharge planning, and home medical care. Promote the long-term care bases in several ways, so that people can understand the pass way to find the bases. In addition to integrating the sanitation and social welfare, long-term care 2.0 also extends the recipients, establishes a standard SOP, builds a shared information platform for care service management and a communication bridge between the hospital and sanitation, and uses the media to promote the people's cognition of long-term care 2.0[9].”.
Due to table 1, number “20” should be changed into “5.7” in line 20, page 3.
Thank you for these observations. We had fixed the presentation style of the article. Instead of “20”, it should be “5.7”.
Number “76.1” should be converted into “67.1” in line 24, page 3.
Thank you for these observations. We had fixed the presentation style of the article. Instead of “76.1%”, it should be “67.1%”.
Number “58.8” should be converted into “58.5” in line 3, page 4.
Thank you for these observations. We had fixed the presentation style of the article. Instead of “58.8”, it should be “58.5”.
The results of univariate analysis can only indicate whether the distribution difference between using long term care group and not using long term care group is statistically significant. In 3.2 section, the description of analysis result should be further modified.
Thank you for these observations. We had completed this description in introduction as the statements, “We analyzed the influenced factors which affected that the patient who had received the discharge planning service utilized long term care 2.0 by Andresen Behavioral Model (Table 2). In the predisposing factors. Based on the age distribution, we could understand that most family was elderly couples whether long-term care used or not. Especially, there were up to 65.7% patients, the mean age 75.1 ± 10.9, using long term care resources, and 71.0% patients, who were older than 65-year-old, used long term care resources after discharge. It revealed that age (p=0.001) was a significant factor to using long term care 2.0. In enabling factors, there were 58.5% patients using long term care resources lived in the area without medical institutions, which disclosed that whether there a medical institution in the place of residence would be significant difference (p=0.004). In need factors, card for person with disability would affect that patient used long term care after discharge or not (p=0.002). In the past history, it showed significant difference of the long term care 2.0 utilization in cerebral vascular diseases/stroke (p=0.002), coronary arterial disease/cardiovascular disease (p=0.003) and diabetes mellitus (p=0.009). According to these results, the significant predictive factors for using the service of long term care included age, medical accessibility, card for person with disability, cerebrovascular disease, coronary or cardiovascular disease and diabetic mellitus. However, there was no significant association between sex, tubal insertion, dementia, osteoarthritis, malignancy or end stage renal disease and the utilization of long term care 2.0.
”.
Word “men” should be changed into “mean” in line 7, page 5.
Thank you for these observations. We had fixed the presentation style of the article. Instead of “men”, it should be “mean”.
In discussion section, the author stated “there is no difference between rural and urban area in our study”. The author need to show the analysis results in result section.
Thank you for these observations. We had deleted the statements, “there is no difference between rural and urban area in our study” due to our clerical error.
In discussion section, the author may need to explain more why these factors could influence the usage of long term care and it would be great to compare with other research.
Thank you for these observations. We had completed this description in discussion sections 4.1, 4.2 and 4.3.

Reviewer 3 Report
- How does this research result contribute to the world? Does long-term care 2.0 applicable to the worldwide?
- The author utilized Andersen Behavioral Model as the framework in this study. However, I did not see any description of this model in the introduction and methods.
- The statistical writing method in the text should be improved. For example, if p value is less than 0.5, there is no need to write out the data.
- Reconsidering the necessity of discussing whether the demographic data is consistent with the literature. It is recommended to discuss the relationship between the characteristics of participants and dependent variables. In addition, it is recommended to describe elderly people's population status in the introduction section.
Author Response
REVIEWER #3
- How does this research result contribute to the world? Does long-term care 2.0 applicable to the worldwide?
Thank you for these observations. We had completed this description in introduction as the statements, “The difference of long-term care between Taiwan and other countries is that we have our own national health insurance, which is a compulsory insurance policy and adopts a nationally consistent rate range. The national health insurance can also provide medical protection for economically disadvantaged groups. According to the global population structure, the number of people who need the service of long-term care will increase year by year.”
- The author utilized Andersen Behavioral Model as the framework in this study. However, I did not see any description of this model in the introduction and methods.
Thank you for these observations. We had completed this description in Methods section 2.2 as the statements, “According to the Andersen Behavioral model, personal medical behavior would be influenced by predisposing factors, enabling factors and need factors [14]. Moreover, in the past decades, Andersen Behavior Model was widely applied in a large amount of literature on the discussion of medical services and other associated factors.”.
- The statistical writing method in the text should be improved. For example, if p value is less than 0.5, there is no need to write out the data.
We agree with the reviewer and revised the article base on your suggestion. We had fixed the presentation style of the article.
- Reconsidering the necessity of discussing whether the demographic data is consistent with the literature. It is recommended to discuss the relationship between the characteristics of participants and dependent variables. In addition, it is recommended to describe elderly people's population status in the introduction section.
Thank you for these observations. We had completed this description in introduction as the statements, “Until January 2021, the population of elder than 65 year-old was 16.15% and elder than 85 year-old was 10.3% (total population in Taiwan was 23,548,633) [1].”

Reviewer 4 Report
Dear Authors,
Your paper deals with an important topic in the field of organisational issues of long-term-care provision for the elderly.
In the following please find my comments on your paper:
Title: needs more specification
Introduction: Needs structure as well as the explanation of several terms (for example “the standard of aged society [3]”) and a comprehensive description of the purpose, target-groups and measures of “Long-term Care Plan 1.0” and “Long-term care 2.0”.
You criticize previous research on discharge planning without describing the identified research gaps which serve as a starting point for your empirical investigation.
Specification of purpose of the study and research questions are missing.
Methods: This section lacks information on sampling and procedure of information gathering. Your study comprises information on 280 patients. Information on the procedure of data collection and the questionnaire is necessary.
The application of Andersen Behavioural Model of Health Care Utilization aims to combine environmental factors and personal attributes in order to draw conclusions on their influence on health behaviour. Your analysis disregards the relevance of environmental factors (housing situation, availability of caregiving relatives, distance-to-clinic) and exclusively includes two demographic characteristics (age and sex) and the clinical picture of the respondents.
Discussion: In this section relevant information on the respondents’ profile (marital status!) is provided. This information is missing in the description of the profile of the patients.
Study limitation: The exclusion of the perspective of the “main caregivers” is an important limitation. (see comments referring to “Methods”).
Conclusion: Too broad referring to contextualisation, more specification needed.
All the best!

Author Response
REVIEWER #4
Dear Authors,
Your paper deals with an important topic in the field of organisational issues of long-term-care provision for the elderly.
In the following please find my comments on your paper:
Title: needs more specification
Thank you for these observations. We had fixed the presentation style of Title. Instead of “Factors Influenced Patients on Using Long Term Care Service of Discharge Planning: A hospital-based cross-sectional study in Eastern Taiwan”, it should be “Factors Influenced Patients on Using Long Term Care Service of Discharge Planning by Andersen Behavioral Model: A hospital-based cross-sectional study in Eastern Taiwan”.
You criticize previous research on discharge planning without describing the identified research gaps which serve as a starting point for your empirical investigation.
Thank you for these observations. We had completed this description in introduction as the statements,” Long-term care in Taiwan begins at discharge planning in 1994, which is a centralized, coordinated and continuous process of collaboration with patient and their family through inter-professional medical teams [6]. Moreover, discharge planning also emphasizes the connection of long-term care resources in the community during hospitalization, which is an important preparation for patients to return to the community. National Ten-year Long-term Care Plan 1.0 (long-term care 1.0) was performed during 2007 to 2016 for promoting local aging, which included the service of respite care at home, home care, home nursing care, home rehabilitation, transportation, assistive device service, nutritional catering and institutional service [7]. However, there were many limitation of long-term care 1.0. Only those people with the following qualifications could use long-term care 1.0: card for people with disability and older than 50-year-old, mountain aboriginal who were older than 55-year-old, the disable and older than 65-year-old elderly, IADL-dependent and older than 65-year-old [8]. The payment of long-term care service was inflexible: the service hour of different item was fixed, and the payment of service could not be cross-project. Besides, the lack of care attendants and over working led to restrictions of service amounts and low retention rate (24%). Therefore, long-term care 1.0 was upgraded to 2.0 since 2017. The served population are expanded to fragile elder older than 65 years old, people younger than 49 years old with card of disability, people older than 50 years old with dementia, and disable aboriginals who are older than 55. After the expansion, the number of service recipients increased from 511,000 to 738,000, with an increase of about 44%[8]. Moreover, the content of service items also expanded from 8 to 17, extending forward to the prevention stage and backward to the terminal stage. The newly included items are: dementia day-care, community integration, caregiver supporting bases, community prevention care, prevention/delay disability, discharge planning, and home medical care. Promote the long-term care bases in several ways, so that people can understand the pass way to find the bases. In addition to integrating the sanitation and social welfare, long-term care 2.0 also extends the recipients, establishes a standard SOP, builds a shared information platform for care service management and a communication bridge between the hospital and sanitation, and uses the media to promote the people's cognition of long-term care 2.0[9].”
Specification of purpose of the study and research questions are missing.
Thank you for these observations. We had completed this description in introduction as the statements,” There were several studies about the discharge planning that could shorten the dura-tion of hospitalization, reduce re-admission rate, lower the medical cost, and raise the medical care quality [12,13,15]. However, the research about the connection between dis-charge planning and long-term care was scanty. Therefore, it is worth to explore the pre-disposing factors that influence the utilization of long-term care [16]. Therefore, in this current study, the purpose were looking forward to disclosing the causes which affect the patients who had already received the service of discharge planning willing to accept the service of long-term care, and promoting the experience of long-term care in Taiwan and share it with other countries.”
Methods: This section lacks information on sampling and procedure of information gathering. Your study comprises information on 280 patients. Information on the procedure of data collection and the questionnaire is necessary.
Thank you for these observations. We had completed this description in Methods section 2.2 as the statements, “In this cross-sectional study, case review was used, and the patients who received the service of discharge planning at the hospital (Taipei Veterans General Hospital Yuli Branch) which was located in the southern Hualien were included during November, 2017 to October, 2018. The number of bed in the hospital was 95. This current study past the audit of the Institutional Review Board, and the trial number was 19-008-C. In addition to receiving the service of discharge planning, the patients who were included were also compatible with the following conditions of long-term care 2.0: persons younger than 49 years old with card of disability, persons older than 50 years old with dementia, disable mountain aboriginals who are older than 55 years old, persons older than 65 years old with activities of daily dependent or persons older than 65 years old living alone with instrumental activities of daily living dependent. 294 patients were included and received the service of discharge planning during the hospitalization. The excluded criteria were expired during the hospitalization, transferred to other hospital, or not compatible with the conditions of long-term care 2.0. However, 14 patients were excluded (6 patients were expired, 5 patients were not suitable with the rule of long term care 2.0, and 3 patients were transferred to other hospital), and there were 280 patients included finally. We asked 280 patients to fill out a brief questionnaire after receiving the service of discharge planning during the hospitalization.”.
The application of Andersen Behavioural Model of Health Care Utilization aims to combine environmental factors and personal attributes in order to draw conclusions on their influence on health behaviour. Your analysis disregards the relevance of environmental factors (housing situation, availability of caregiving relatives, distance-to-clinic) and exclusively includes two demographic characteristics (age and sex) and the clinical picture of the respondents.
Thank you for these observations. We had completed this description in Methods section 2.2 as the statements, “According to the Andersen Behavioral Model, personal medical behavior would be influenced by predisposing factors, enabling factors and need factors [17]. Moreover, in the past decades, Andersen Behavior Model was widely applied in a large amount of literature on the discussion of medical services and other associated factors. Therefore, we setup our structure of current study based on the Andersen Behavioral Model [17]. We collected data according to reasons for disabled elderly need care in Taiwan, Japan and USA, and separated it into predisposing factors, enabling factors and need factors [18-20]. Predisposing factors included age and sex; enabling factors included medical accessibility (the area in our study is narrow and long, the length around 100km, and the medical source is scanty); need factors included tubal insertion(nasogastric tube, Foley tube, tracheostomy, colostomy...etc.), card for person with disability, chronic diseases (cerebrovascular disease, dementia, osteoarthritis, coronary/ cardio-vascular disease, chronic obstructive pulmonary disease/asthma, malignancy, end stage renal disease, diabetic mellitus). From November 2017 to October 2018, in Taipei Veterans General Hospital Yuli Branch, we recruited 280 valid patients who compatible with the conditions of long-term care 2.0and invited them for a standardized interview.
”.
Discussion: In this section relevant information on the respondents’ profile (marital status!) is provided. This information is missing in the description of the profile of the patients.
Thank you for these observations. All of our research subjects were married, so there were no analysis results to compare with other research.
Study limitation: The exclusion of the perspective of the “main caregivers” is an important limitation. (see comments referring to “Methods”).
Thank you for these observations. We had completed this description in Study limitation.
Conclusion: Too broad referring to contextualisation, more specification needed.
Thank you for these observations. We had completed this description in Conclusion.
All the best!

Round 2
Reviewer 4 Report
Dear Authors,
You took into account all of my recommendations but one: the description of the questionnaire. Please, add the questionnaire.
Yours sincerely
